# Implicit Neural Network for Dynamic Graphs

## Abstract

Recent works have demonstrated that graph convolution neural networks fail either to capture long-range dependencies in the network or suffer from over-smoothing issues. Several recent works have proposed implicit graph neural networks to remedy the issues. However, despite these issues being magnified in dynamic graphs, where the feature aggregation occurs through both the graph neighborhood and across time stamps, no prior work has developed implicit models to overcome these issues. Here we present IDGNN, a novel implicit neural network for dynamic graphs. We demonstrate that IDGNN is well-posed, i.e., it has a unique fixed point solution. However, the standard iterative algorithm often used to train implicit models is computationally expensive in our setting and cannot be used to train IDGNN efficiently. To overcome this, we pose an equivalent bi-level optimization problem and propose a single-loop training algorithm. We conduct extensive experiments on real-world datasets on both classification and regression tasks to demonstrate the superiority of our approach over the state-of-the-art baseline approaches. We also demonstrate that our bi-level optimization framework maintains the performance of standard iterative algorithm while obtaining up to **1600x** speed-up.

## 1 Introduction

Graph Convolution Network (GCN) (Kipf & Welling, 2016) and its subsequent variants (Veličković et al., 2018; Li et al., 2018b) have raised the bar in predictive tasks in various applications: molecular prediction (Park et al., 2022), recommendation (Liao et al., 2022), and hyperspectral image classification (Hong et al., 2020). GCNs have also been extended to the dynamic setting, where the graph changes over time. Even in the dynamic setting, GCNs have achieved state-of-the-art results for tasks including rumor detection (Sun et al., 2022) and traffic prediction (Li et al., 2023).

Despite their success, GCNs have a significant drawback. Empirical evidence suggests that deepening the layers of GCN can lead to a notable decline in their performance, even beyond a few (2-4) layers. This phenomenon is called over-smoothing (Li et al., 2018a), wherein the stacked GCN layers gradually smooth out the node-level features, resulting in a degradation of performance on node-level tasks. Meanwhile, to capture long-range dependencies, multiple GCN layers need to be stacked since a single GCN layer can only aggregate information from neighboring nodes that are one hop away. This creates a conflict where, on the one hand, one would like to capture dependencies between nodes that are far away in the network by stacking multiple layers of GCN together. On the other hand, one would like to avoid the over-smoothing problem by only using a few layers. To tackle this dilemma in the static setting, Gu et al. (2020) proposed an implicit graph neural network (IGNN), which iterates the graph convolution operator until the learned node representations converge to a *fixed-point* representation. Since there is no a priori limitation on the number of layers, it is able to alleviate the over-smoothing problem without sacrificing long-range dependency.

In the case of dynamic graphs, GCN needs to aggregate information over the current graph topology and historical graphs to learn meaningful representations. This corresponds to stacking at least one GCN layer per time stamp. This is exacerbated in current practice, where widely used approaches stack multiple GCN layers even within a single time stamp. Therefore, the over-smoothing issue is even magnified for GCNs in dynamic settings. However, very few prior works study the over-smoothing phenomenon in dynamic graphs: Yang et al. (2020) proposes an L2 feature normalization process to alleviate the over-smoothing in dynamic graphs and Wang et al. (2022) mitigates the over-smoothing problem by emphasizing the importance of low-order neighbors via a node-wise encoder.

However, these approaches either rescale features or forget neighborhood information, both of which are not ideal.

Inspired by the success of implicit GCNs in overcoming the over-smoothing and long-range dependency conflict, here we propose to develop an implicit neural network for dynamic graphs from the first principle. However, we noticed that there are multiple barriers including *i)* determining if fixed-point (converged) representations exist in dynamic graphs; *ii)* and if yes, efficiently training a model to find these fixed-point representations. In this paper, we overcome barrier *i)* by first proving the existence of the fixed-point representations on periodic dynamic graphs and extending this result to design an implicit model for general dynamic graphs and barrier *ii)* by designing an efficient bilevel optimization algorithm. The key contributions of the paper are as follows:

- We propose an effective embedding learning framework for dynamic graphs. To the best of our knowledge, IDGNN is the first method that tackles the dynamic graph problem via an implicit neural network.

- We present a bilevel optimization viewpoint of our method and propose a novel stochastic optimization algorithm that can efficiently train our model. We conducted an ablation study to show that our proposed optimization algorithm is faster than naive gradient descent up to 1600 times.

- We conduct comprehensive comparisons with existing methods and demonstrate that our method alleviates the over-smoothing problem and outperforms the state-of-the-art Temporal GNN models on both classification and regression tasks.

## 2 RELATED WORK

**Dynamic Graph Representation Learning:** GNN has been successful for static graphs, leading to the development of GNN-based algorithms for dynamic graphs (Khoshraftar & An, 2022). DyGNN (Ma et al., 2020) comprises two components: propagation and update, which enable information aggregation and propagation for new interactions. EvolveGCN (Pareja et al., 2020) uses an RNN to update GCN parameters and capture dynamic graph properties. Sankar et al. (2020) proposes a Dynamic Self-Attention Network (DySAT) with structural and temporal blocks to capture graph information. TGN (Rossi et al., 2020) models edge streaming to learn node embeddings using an LSTM for event memory. TGAT (Xu et al., 2020) considers the time ordering of node neighbors. Gao & Ribeiro (2022) explores the expressiveness of temporal GNN models and introduces a time-then-graph framework for dynamic graph learning, leveraging expressive sequence representations like RNN and transformers.

**Implicit Graph Models:** The implicit models or deep equilibrium models are models with implicitly determined parameters. Bai et al. (2019). propose an equilibrium model for sequence data based on the fixed-point solution of an equilibrium equation. El Ghaoui et al. (2021) introduce a general implicit deep learning framework and discuss the well-posedness of implicit models. Gu et al. (2020) demonstrate the potential of implicit models in graph representation learning, specifically with their implicit model called IGNN, which leverages a few layers of graph convolution network (GCN) to discover long-range dependencies. Park et al. (2021) introduces the equilibrium GNN-based model with a linear transition map, and they ensure the transition map is contracting such that the fixed point exists and is unique. Liu et al. (2021) propose an infinite-depth GNN that captures long-range dependencies in the graph while avoiding iterative solvers by deriving a closed-form solution. Chen et al. (2022) employ the diffusion equation as the equilibrium equation and solve a convex optimization problem to find the fixed point in their model.

**Implicit Models Training:** Efficiently training implicit models has always been a key challenge. Normally, the gradient of implicit models is obtained by solving an equilibrium equation using fixed-point iteration or reversing the Jacobian matrix (Gu et al., 2020). However, training implicit models via implicit deferential introduces more computational overhead; the following works aim to reduce the training cost. Geng et al. (2021) propose phantom gradient to accelerate the training of implicit models based on the damped unrolling and Neumann series. Li et al. (2022) leverage stochastic proximal gradient descent and its variance-reduced version to accelerate the training.

## 3 METHODOLOGY

Here, we first consider a discrete-time cyclic dynamic graphs $\mathcal{G} = \{G_1, ..., G_T\}$ where each $G_t$ is a snapshot graph at time $t$ represented by the tuple $(A_t, X_t)$. $A_t \in \mathbb{R}^{n \times n}$ is the adjacency matrix of $G_t$ and $X_t \in \mathbb{R}^{l \times n}$ is the node attribute matrix, where $n$ the number of distinct nodes in all snapshots and $d$ the dimension of node attribute. In general, implicit models have the following framework,

$$Z^{k+1} = f(Z^k, X) \tag{1}$$

where $f$ is neural network, $X$ is data and $Z$ is learned representation. When we stack infinite layers of $f$, we obtain the fixed-point representation $Z^* = \lim_{k \to \infty} Z^{k+1} = \lim_{k \to \infty} f(Z^k, X) = f(Z^*, X)$. Thus, the key to designing an implicit model for dynamic graphs is to provide a function $f$ with a convergence guarantee.

### 3.1 IMPLICIT MODEL FOR DYNAMIC GRAPHS

We construct the following building block for dynamic graphs with $T$ time stamps:

$$Z_2^{k+1} = \sigma(W_2 Z_1^k A_2 + V X_2)$$
$$\cdots$$
$$Z_T^{k+1} = \sigma(W_T Z_{T-1}^k A_T + V X_T)$$
$$Z_1^{k+1} = \sigma(W_1 Z_T^k A_1 + V X_1) \tag{2}$$

In the model presented above, the learned representations $Z_2^{k+1}$ of the nodes in the second time stamp in the $(k+1)^{th}$ layer depend on the embeddings $Z_1^k$ of nodes in the first time stamp learned in $k^{th}$ layer and the feature vector in the second time stamp $X_2$. This design enables us to propagate information among time stamps when stacking layers. The parameters for the $t$-th layer of the model are denoted as $W_t \in \mathbb{R}^{d \times d}$ and $V \in \mathbb{R}^{d \times l}$ (with $V$ being a shared weight). Note that our theory still holds when $V$ is not shared. We opt for a shared $V$ for simplicity, and a thorough discussion on this choice is presented in the Appendix. Following the principle of the implicit model (El Ghaoui et al., 2021; Bai et al., 2019; Gu et al., 2020), we apply our model iteratively until convergence. We consider the converged result $\{Z_1, \ldots, Z_T\}$ as the final embeddings. Consequently, the final embeddings have to satisfy the system of equations in (2) and can be considered a fixed point solution to (2). However, at this point, it is not clear whether it always exists for arbitrary graph $\mathcal{G}$.

*Well-posedness* is a property that an implicit function, such as in (2), possesses a unique fixed point solution. We note that Gu et al. (2020) has already established the well-posedness for one layer implicit graph neural network on static graphs as given by the following Lemma.

**Lemma 1** *The equilibrium equation $z = \sigma(Mz + b)$ has a unique fixed point solution if $|||M|||_{op} < 1$, where $\|.\|_{op}$ is the operator norm, and $\sigma(\cdot)$ is an element-wise non-expansive function.*

In order to establish the well-posedness result for our model, we first introduce a vectorized version of our model and leverage Lemma 1. The vectorized version of Equation (2) is as follows.

$$z_2^{k+1} = \sigma(M_2 z_1^k + \mathbf{vec}(V X_2))$$
$$\cdots$$
$$z_T^{k+1} = \sigma(M_T z_{T-1}^k + \mathbf{vec}(V X_T))$$
$$z_1^{k+1} = \sigma(M_1 z_T^k + \mathbf{vec}(V X_1)) \tag{3}$$

where $z = \mathbf{vec}(Z)$ and $M_i = A_i^\top \otimes W_i$, and $\otimes$ is the Kronecker product. Note that 3 can also be expressed using a single matrix. This transformation involves sequentially connecting the shared nodes between the graphs. Thus, the formula (3) can be reformulated as follows:

$$\begin{bmatrix} z_1 \\ z_2 \\ z_3 \\ \vdots \\ z_T \end{bmatrix} = \sigma \left( \begin{bmatrix} 0 & 0 & \cdots & 0 & M_1 \\ M_2 & 0 & \cdots & 0 & 0 \\ 0 & M_3 & \cdots & 0 & 0 \\ \vdots & \vdots & \ddots & \vdots & \vdots \\ 0 & 0 & \cdots & M_T & 0 \end{bmatrix} \begin{bmatrix} z_1 \\ z_2 \\ z_3 \\ \vdots \\ z_T \end{bmatrix} + \begin{bmatrix} \mathbf{vec}(V X_1) \\ \mathbf{vec}(V X_2) \\ \mathbf{vec}(V X_3) \\ \vdots \\ \mathbf{vec}(V X_T) \end{bmatrix} \right) \tag{4}$$

We omit the superscript for simplicity. It is evident that equations (4) and (3) are equivalent, with Formula (4) representing a single equilibrium equation. Note that Equation (4) represents the time-expanded static view of our original dynamic graph $\mathcal{G}$. As a result, we can readily deduce the well-posedness result of (3) based on Lemma (1).

**Theorem 1** *For element-wise non-expansive function $\sigma(\cdot)$, the coupled equilibrium equations (3) have a unique fixed point solution if $\|\|\mathcal{M}\|\|_{op} < 1$, where $\mathcal{M}$ define as*

$$
\begin{bmatrix}
0 & \cdots & 0 & M_1 \\
M_2 & \cdots & 0 & 0 \\
\vdots & \vdots & \ddots & \vdots \\
0 & \cdots & M_T & 0
\end{bmatrix}
$$

*and $\|\|M\|\|_{op}$ is the operator norm of $M$, which is the largest absolute eigenvalue. Furthermore, this means $\|\|M_t\|\|_{op} < 1$ for any $t = 1, ..., T$.*

In order to maintain $\|\mathcal{M}\|_{op} < 1$, it is necessary to ensure that the condition $\lambda_{\mathrm{pr}}(|W|)\lambda_{\mathrm{pr}}(A) < 1$ is satisfied, where $\lambda_{\mathrm{pr}}(\cdot)$ represents the Perron-Frobenius eigenvalue. However, guaranteeing this constraint can be challenging in general. To overcome this challenge and ensure the condition $\|\|\mathcal{M}\|\|_{op} < 1$, we can leverage the following theorem, which imposes a more stringent requirement on $W$. Following the approach used in Gu et al. (2020), we can utilize a convex projection to ensure the satisfaction of $W$.

**Theorem 2** *Let $\sigma$ be an element-wise non-expansive non-linear function. The coupled equilibrium equations satisfy the well-posedness condition, namely $\|\|W\|\|_{op}\|A\|_{op} < 1$. There exists rescale coupled equilibrium equations, which satisfy $\|W\|_\infty \|A\|_{op} < 1$, and the solutions of these two equations are equivalent.*

## 4 TRAINING

The key challenges in training our model lie in determining how to perform backpropagation effectively, especially within the context of implicit models. To address this challenge, most implicit models rely on estimating gradients using the Implicit Function Theorem (Bai et al., 2019; Gu et al., 2020). This approach offers several advantages, such as eliminating the need to store intermediate results during the forward pass and enabling direct backpropagation through the equilibrium point.

We first explore the naive gradient descent method employing the Implicit Function Theorem. While widely used in various techniques, this approach presents certain drawbacks when applied to our specific model, particularly in terms of computational overhead. Subsequently, we introduce an efficient training algorithm for our model, which adopts a bilevel viewpoint of our problem. This novel approach allows us to overcome the limitations of the naive gradient descent method, resulting in improved computational efficiency during training.

**Objective:** Let us consider classification and regression tasks for the following discussion. We consider a dataset $(\mathcal{G}_i, y_i)_{i=1}^N$, which consists of $N$ dynamic graphs and their corresponding targets. Each dynamic graph comprises $T$ time stamps. We utilize a neural network, parameterized by $\theta$ and denoted as $f_\theta(.)$, to map graph embeddings to their respective targets. Our objective can be summarized as follows:

$$
\min_{\theta, \boldsymbol{W}, V} \mathcal{L}(\theta, \boldsymbol{W}, V) = \sum_{i=1}^N \ell(f_\theta(z_T^{(i)}), y_i) \tag{5}
$$

$$
\text{s.t. } z_2^{(i)} = \sigma\left( (A_2^{(i)} \otimes W_2^\top) z_1^{(i)} + \mathbf{vec}(V X_2^{(i)}) \right)
$$

$$
\cdots
$$

$$
z_1^{(i)} = \sigma\left( (A_1^{(i)} \otimes W_1^\top) z_T^{(i)} + \mathbf{vec}(V X_1^{(i)}) \right),
$$

$$
\|W_t\|_\infty \le \frac{\kappa}{\|A_t^{(i)}\|_\infty}, i = 1, ..., N, t = 1, ..., T
$$

where $\ell$ is a loss function ( e.g. cross entropy loss, mean square error). Let $A_j^{(i)}$, $X_j^{(i)}$, and $z_j^{(i)}$

represent the adjacency matrix, feature, and embedding, respectively, of the $j$-th timestamp within the $i$-th dynamic graph.

**Naive Gradient Descent:** Naive Gradient Descent operates in a straightforward manner: it first finds the fixed-point embedding through fixed-point iteration and then computes the gradient based on this embedding. The gradient with respect to parameter $\theta$ can be obtained as $\frac{\partial \mathcal{L}}{\partial \theta}$, which is easily computed using autograd functions given the fixed point. However, computing the gradient for other parameters presents a greater challenge. Let $\frac{\partial \mathcal{L}}{\partial P_i}$ represent the gradient with respect to $W_i$ or $V_i$. For simplicity, we only consider the gradient of only one dynamic graph. The gradient is computed as $\frac{\partial \mathcal{L}}{\partial P_i} = \sum_{j=1}^{T} \frac{\partial \mathcal{L}}{\partial z_j} \frac{\partial z_j}{\partial P_i}$. The computation of $\frac{\partial \mathcal{L}}{\partial z_j}$ can be achieved through the autograd mechanism. However, determining $\frac{\partial z_j}{\partial P_i}$ is non-trivial due to the implicit definition of $z_j$.

To avoid using tensors, we represent the matrices $W$, and $V$ as column-wise vectorized vectors denoted as $w$, and $v$, respectively. The learned embeddings must satisfy the following equations:

$$F_2(\boldsymbol{z}, W, V) = z_2 - \sigma(M_2 z_1 + \mathbf{vec}(V X_2)) = 0$$

$$\vdots$$

$$F_T(\boldsymbol{z}, W, V) = z_T - \sigma(M_T z_{T-1} + \mathbf{vec}(V X_T)) = 0$$
$$F_1(\boldsymbol{z}, W, V) = z_1 - \sigma(M_1 z_T + \mathbf{vec}(V X_1)) = 0$$

According to the implicit function theorem, we can calculate the gradients $\frac{\partial z}{\partial w}$ and $\frac{\partial z}{\partial v}$ through implicit differentiation. The detailed derivation is provided in the Appendix. Therefore, for any time stamp $k$, the gradient of $z_k$ with respect to the $a$-th layer of GCN, $w_a$, can be expressed as:

$$\frac{\partial z_k}{\partial w_a} - \Sigma'_k \odot \left( \delta_{ak} H_k \otimes I + M_k \frac{\partial z_{k-1}}{\partial w_a} \right) = 0 \tag{6}$$

Here, $H_k := (Z_{k-1} A_k)^\top$, $\delta_{ak}$ is the indicator function which equals 1 only when $a = k$, and $\odot$ denotes element-wise multiplication. Each column of $\Sigma'_k$ represents the vectorized $\sigma'(M_k z_{k-1} + \mathbf{vec}(V X_k))$, where $\sigma'(\cdot)$ is the derivative of $\sigma(\cdot)$. Similarly, we can compute the gradient of $z_k$ with respect to $v$.

$$\frac{\partial z_k}{\partial v} - \Sigma'_k \odot \left( X_k^\top \otimes I + M_k \frac{\partial z_{k-1}}{\partial v} \right) = 0 \tag{7}$$

**Per-iteration Complexity of naive gradient descent:** Formulas (6) and (7) reveal that the derivatives are determined by equilibrium equations. Consequently, in order to compute the derivatives, we must solve these equations using fixed-point iteration. Each layer necessitates one round of fixed-point iteration, and in total, including $V$, we need to perform fixed-point iteration $T + 1$ times. The major computational overhead arises from the multiplication of $M$ with the derivatives, resulting in a complexity of $O((nd)^2 d^2)$. Each fixed-point iteration involves $T$ instances of such computations. Consequently, the overall runtime for each update is $O(T^2 n^2 d^4)$. Although the adjacency matrix is sparse, it only reduces the complexity to $O(T^2 n d^4)$. This limitation in complexity poses constraints on applying our model to deeper dynamic graphs and hampers our ability to utilize large embeddings.

### 4.1 Efficiently Update via Bilevel Optimization

To address the previously mentioned challenges, we can turn to Bilevel Optimization as a potential solution, considering that Formula (5) can be regarded as a conventional problem in bilevel optimization. To facilitate this approach, we will rephrase Formula (5) using the subsequent Lemma.

**Lemma 2** *If Formula (3) has an unique embedding $\{z_1^*, \cdots, z_T^*\}$, let $\bar{j} := j \mod T$, then the equation $z_j = \sigma(M_{\overline{j+T}} \sigma(M_{\overline{j+T-1}} \cdots \sigma(M_{\overline{j+1}} z_j + \mathbf{vec}(V X_{\overline{j+1}})) \cdots + \mathbf{vec}(V X_{\overline{j+T-1}})) + \mathbf{vec}(V X_{\overline{j+T}}))$ has the same fixed point as $z_j^*$ for any $j \in \{1, \cdots, T\}$, and vice versa.*

According to Lemma 2, we can convert the problem presented in Equation (5) into a standard bilevel optimization problem. This transformation allows us to utilize established techniques and method-

---

**Algorithm 1** Stochastic Training Algorithm for IGDNN

---

**Require:** $\mathcal{D} = \{(\mathcal{G}_i, y_i)\}_{i=1}^N, \eta_1, \eta_2, \gamma$
**Ensure:** $\omega, \theta$
    **for** $t = 0, 1, ..., M$ **do**
        Sample a batch data $\mathcal{B} \in \mathcal{D}$
        $z_j^{t+1} = \begin{cases} (I - \eta_1)z_j^t + \eta_1 \phi(z_j^t, \omega^t; \mathcal{G}_i) & j \in \mathcal{B} \\ z_j^t & \text{o.w.} \end{cases}$
        $v_j^{t+1} = \begin{cases} (I - \eta_2 \nabla_{zz}^2 g(z, \omega^t))v_j^t + \eta_2 \nabla_z \ell_j(z_j^t, \omega^t) & j \in \mathcal{B} \\ v_j^t & \text{o.w.} \end{cases}$
        Update gradient estimator

$$\Delta^{t+1} = \frac{1}{|\mathcal{B}|} \sum_{j \in \mathcal{B}} \left[ \nabla_\omega \ell_j(z_j^t, \omega^t) - \nabla_{\omega z}^2 g_j(z_j, \omega^t) v_j^t \right]$$

    $m^{t+1} = (1 - \gamma)m^t + \gamma \Delta^{t+1}$
    $\omega^{t+1} = \Pi_\Omega \left( \omega^t - \eta_0 m^{t+1} \right)$
    **end for**

---

ologies for solving such problems.

$$\min_{\theta, \boldsymbol{W}, V} \mathcal{L}(\theta, \boldsymbol{W}, V) = \sum_{i=1}^N \ell(f_\theta(z^{(i)}), y_i) \tag{8}$$

$$\text{s.t. } z^{(i)} = \arg\min_z \|z - \phi(z, \boldsymbol{W}, V; \mathcal{G}_i)\|_2^2$$

$$\|W_t\|_\infty \le \frac{\kappa}{\|A_t^{(i)}\|_\infty}, i = 1, ..., N, t = 1, ..., T$$

where $\phi(z, \boldsymbol{W}, V; \mathcal{G}_i) = \sigma(M_T^{(i)}...\sigma(M_1^{(i)}z + VX_1^{(i)})... + \mathbf{vec}(VX_T^{(i)}))$, $\kappa \in (0, 1]$ is the hyperparameter to control the strength of the projection. The main difference between these problems lies in the constraint. Formula (8) introduces explicit constraints solely on the last snapshot, leading to a multi-block bilevel optimization problem. This problem has been investigated in recent studies by Qiu et al. (2022) and Hu et al. (2022). Qiu et al. (2022) focus on top-K NDCG optimization, formulating it as a compositional bilevel optimization with a multi-block structure. Their approach simplifies updates by sampling a single block batch in each iteration and only updating the sampled blocks. Hu et al. (2022) employ a similar technique but address a broader range of multi-block min-max bilevel problems.

However, these state-of-the-art bilevel optimization algorithms are designed to address problems with strongly convex lower problems, which does not hold true for our problem. For simplicity, we use notation $\omega = \{\boldsymbol{W}, V\}$, let $g_i(z, \omega)$ represent the $i$th-block lower problem, defined as $\|z - \phi(z, \omega; \mathcal{G}_i)\|_2^2$, and let $\ell_i(z, \omega) := \ell(f_\theta(z), y_i)$. It is evident that $g(.)$ is nonconvex with respect to $z$. Additionally, these methods utilize stochastic gradient descent on the lower level in each iteration, leading to potential extra computation. Nevertheless, it is crucial to note that the optimal solution to our lower-level problem corresponds to the fixed point of Eq (3), as per Lemma 2. Leveraging this insight, we employ a fixed-point iteration to update the lower-level solution. We propose a single loop algorithm 1 with fixed-point updates.

To better illustrate our algorithm, let us introduce the hypergradient with respect to $\omega$ as follows:

$$\nabla \mathcal{L}(\omega) = \frac{1}{N} \sum_{i=1}^N \nabla \ell_i(z^{(i)}, \omega) - \nabla_{\omega z}^2 g_i(z^{(i)}, \omega) \left[ \nabla_{zz}^2 g_i(z^{(i)}, \omega) \right]^{-1} \nabla_z \ell_i(z^{(i)}, \omega)$$

If we compute this directly, we may encounter problems with the Hessian $\left[ \nabla_{zz}^2 g_i(z^{(i)}, \omega) \right]^{-1}$ for each block. Inspired by Hu et al. (2022) and Qiu et al. (2022), we approximate $\left[ \nabla_{zz}^2 g_i(z^{(i)}, \omega) \right]^{-1} \nabla_z \ell_i(z^{(i)}, \omega)$ using $v_i$ for each block by moving average estimation.

Table 1: Statistics of datasets. $N$: number of dynamic graphs, $|V|$: number of nodes, $\min |E_t|$: minimum number of edges, $\max |E_t|$: maximum number of edges, $T$: window size, $d$: feature dimension, $y$ label dimension

|  | $N$ | $|V|$ | $\min |E_t|$ | $\max |E_t|$ | $T$ | $d$ | $y$ |
|---|---|---|---|---|---|---|---|
| Brain10 | 1 | 5000 | 154094 | 167944 | 12 | 20 | 10 |
| DBLP5 | 1 | 6606 | 2912 | 5002 | 10 | 100 | 5 |
| Reddit4 | 1 | 8291 | 12886 | 56098 | 10 | 20 | 4 |
| PeMS04 | 16980 | 307 | 680 | 680 | 12 | 5 | 3 |
| PeMS08 | 17844 | 170 | 548 | 548 | 12 | 5 | 3 |
| English-COVID | 54 | 129 | 836 | 2158 | 7 | 1 | 1 |

More specifically, we maintain a $v_i$ to track the optimal point of the following problem $\min_v \frac{1}{2} v^T \nabla^2_{zz} g_i(z^{(i)}, \omega) v - v^T \nabla_z \ell_i(z^{(i)}, \omega)$ for each block. Let $z_i$ be a moving average approximation to the optimal lower-level solution $z^{(i)}$. Moreover, we use fixed-point iteration to update $z$, as presented in Algo. 1. We do not want to update all blocks in every iteration since this is impractical when the number of blocks is large. To address this issue, we use stochastic training. For sampled blocks, we update their $z$ and $v$, and we compute the hypergradient. In cases where the lower-level problem is strongly convex, the errors introduced by these approximations are well-contained (Hu et al., 2022). We notice our lower-level problem admits a unique fixed point, and then employing fixed-point iteration becomes an efficient means of attaining the optimal lower-level solution, akin to the effectiveness of gradient descent under strong convexity. Hence, it is justifiable to assert that our approximations are effective in this scenario, with empirical evidence robustly endorsing their practical efficacy.

It is important to note that the multiplication $\nabla^2_{\omega z} g_j(z_j, \omega^t) v_j^t$ can be efficiently obtained using a trick called Hessian vector product. As a result, the training time for our algorithm is proportional to normal backpropagation, eliminating the need for fixed-point iterations.

**Per-iteration Complexity of naive gradient descent:** the main computational overheads are updating v and estimating gradient. Both steps are involved with estimating a huge Hassian matrix, but, in practice, we can use a trick called Hassian-Vector-Product to avoid explicitly computing the Hessian matrix. Therefore, the dominant runtime of bi-level optimization is three times backpropagation. Each backpropagation takes $O(Tnd^2 + Tn^2 d)$.

## 5 EXPERIMENTS

In this section, we present the performance of IDGNN in various tasks, including effectiveness in capturing long-range dependencies and avoiding over-smoothing on a synthetic dataset. We evaluate the performance of IDGNN against nine state-of-the-art baselines on various real-world datasets. Detailed descriptions of the baseline approaches are presented in the Appendix. Specifically, we perform experiments on three node classification datasets and four node regression datasets. The statistics of the dataset are presented in Table (1). For more detailed descriptions, experimental setup, and hyper-parameter selection, please refer to the Appendix.

### 5.1 OVER-SMOOTHING AND LONG-RANGE DEPENDENCY ON TOY DATA

The toy example aims to test the ability of all approaches to capture long-range dependencies while preventing over-smoothing. The toy data we constructed consists of $\{5, 10, 15, 20\}$ snapshots, with each snapshot being a clique of 10 nodes. Each node has 10 associated attributes. The task is to classify nodes at the last snapshot, where each node represents its own class (i.e., there are a total of 10 classes). The node attributes consist of randomly generated numbers, except for the first snapshot, which uses the one-hot representation of the class. Successful classification of this dataset requires effective information aggregation starting in the initial time stamp, propagating the class label information over time, and avoiding over-smoothing as the information is propagated. In this dataset, there are no testing nodes; all nodes are used for training.

The training results are presented on the right. Our model is compared with GCN-GRU (Seo et al., 2018) and TGCN (Zhao et al., 2019). Based on these models, we propose two more modified

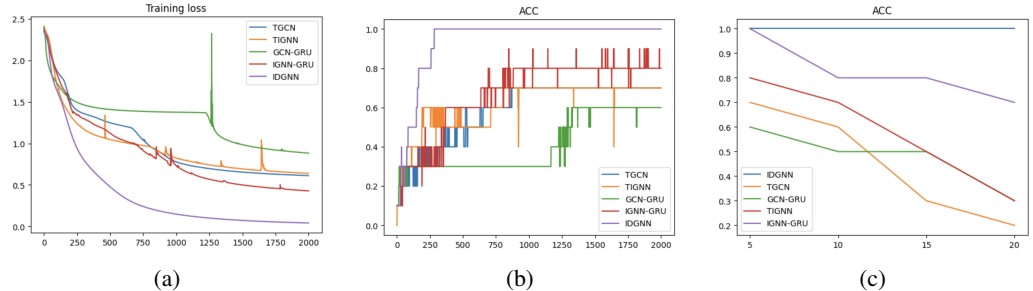

(a)                (b)                (c)

Figure 1: (a) and (b) are training loss and accuracy curves when using 10 layers. The x-axis is epochs, and the y-axis is cross entropy loss and accuracy, respectively. (c) represent the accuracy results of all baselines under different layer settings.

baselines: IGNN-GRU and TIGNN, which are obtained by replacing GCN with IGNN. We ensure the comparison is fair by ensuring a similar number of parameters are used, and we test all models on {5, 10, 15, 20} layers. All methods are trained for a maximum of 2000 epochs, followed by the hyper-parameter selection approach described in the Appendix. As shown in the figure, our method achieves fast convergence to a low loss state and maintains 100% accuracy. In contrast, the baselines fail to reach 100% accuracy due to over-smoothing issues. This demonstrates that our model effectively mitigates the problem of over-smoothing while capturing long-range dependency.

## 5.2 REGRESSION

For node-level tasks, there are two evaluation paradigms: transductive and inductive. Transductive evaluation allows the model to have access to the nodes and edges in testing data during training (i.e., only the labels are hidden), while inductive evaluation involves testing the model on new nodes and edges. In simpler terms, transductive evaluation separates training and testing sets based on nodes, while inductive evaluation separates them based on time stamps (i.e., models are trained on past time stamps and tested on the future). Here, we conduct experiments on both settings.

The datasets we used for the regression task are England-COVID, PeMS04, and PeMS08. We use the mean average percentage error (MAPE) as our evaluation metric. The results are reported in Table 2 with mean MAPE and standard deviation. Our proposed method outperforms other methods in both transductive and inductive settings, with the exception of the inductive case in England-COVID. Our method demonstrates significant improvement for PeMS04 and PeMS08, particularly in the transductive learning scenario. In comparison to the second-best method, our proposed model reduces the error by over 1%, but our model on the inductive learning scenario doesn't enjoy such improvement. We attribute this difference to our model's tendency to separate nodes, even when they have the same labels and topology. We delve into this phenomenon in the Appendix.

Table 2: Performance for classification task (ROCAUC) and regression task (MAPE (%)). Performances on Brain10, England-COVID, PeMS04, and PeMS08 for baseline methods are taken from Gao & Ribeiro (2022). The best performance for each dataset is highlighted in bold, while the second-best performance is underlined.

| | Classification | | | Regression | | | | | |
|---|---|---|---|---|---|---|---|---|---|
| Model | Brain10 | DBLP5 | Reddit4 | England-COVID | | PeMS04 | | PeMS08 | |
| | | | | Trans. | Induc. | Trans. | Induc. | Trans. | Induc. |
| EvolveGCN-O | 0.58±0.10 | 0.639±0.207 | 0.513±0.008 | 4.07±0.73% | 3.88±0.47% | 3.20±0.25% | 2.61±0.42% | 2.65±0.12% | 2.40±0.27% |
| EvolveGCN-H | 0.60±0.11 | 0.510±0.013 | 0.508±0.008 | 4.14±1.14% | 3.50±0.42% | 3.34±0.14% | 2.84±0.31% | 2.81±0.28% | 2.81±0.23% |
| GCN-GRU | 0.87±0.07 | 0.878±0.017 | 0.513±0.010 | 3.56±0.26% | 2.97±0.34% | 1.60±0.14% | 1.28±0.04% | 1.40±0.26% | 1.07±0.03% |
| DySAT-H | 0.77±0.07 | **0.917±0.007** | 0.508±0.003 | 3.67±0.15% | 3.32±0.76% | 1.86±0.08% | 1.58±0.08% | 1.49±0.08% | 1.34±0.03% |
| GCRN-M2 | 0.77±0.04 | 0.894±0.009 | 0.546±0.020 | 3.85±0.39% | 3.37±0.27% | 1.70±0.20% | 1.20±0.06% | 1.30±0.17% | 1.07±0.03% |
| DCRNN | 0.84±0.02 | 0.904±0.013 | 0.535±0.007 | 3.58±0.53% | 3.09±0.24% | 1.67±0.19% | 1.27±0.06% | 1.32±0.19% | 1.07±0.03% |
| TGAT | 0.80±0.03 | 0.895±0.003 | 0.510±0.011 | 5.44±0.46% | 5.13±0.26% | 3.11±0.50% | 2.25±0.27% | 2.66±0.27% | 2.34±0.19% |
| TGN | 0.91±0.03 | 0.887±0.004 | 0.521±0.010 | 4.15±0.81% | 3.17±0.23% | 1.79±0.21% | 1.19±0.07% | 1.49±0.26% | 0.99±0.06% |
| GRU-GCN | 0.91±0.03 | 0.906±0.008 | 0.525±0.006 | 3.41±0.28% | **2.87±0.19%** | 1.61±0.35% | 1.13±0.05% | 1.27±0.21% | 0.89±0.07% |
| IDGNN | **0.94±0.01** | 0.907±0.005 | **0.556±0.017** | **2.65±0.25%** | 3.05±0.25% | **0.53±0.05%** | **0.63±0.04%** | **0.45±0.11%** | **0.50±0.05%** |

(a) Memory and runtime comparison results for all methods on reddit4 and DBLP5 datasets. We report the memory usage using MB and runtime using seconds per window.

| | Reddit4 | | DBLP5 | |
|---|---|---|---|---|
| | Mem. | Time | Mem. | Time |
| EvolveGCN-O | **42** | 0.0649±0.0165 | **52** | 0.0672±0.0144 |
| EvolveGCN-H | 52 | 0.0904±0.0195 | 82 | 0.0997±0.0367 |
| GCN-GRU | 221 | 0.0733±0.0118 | 200 | 0.1142±0.0446 |
| DySAT-H | 181 | 0.1613±0.0555 | 165 | 0.1343±0.0123 |
| GCRN-M2 | 322 | 0.4345±0.0804 | 319 | 0.4934±0.0763 |
| DCRNN | 223 | 0.1697±0.0185 | 278 | 0.2121±0.0397 |
| TGAT | 793 | 0.0750±0.0142 | 338 | 0.0770±0.0150 |
| TGN | 450 | 0.0417±0.0042 | 233 | 0.0454±0.0121 |
| GRU-GCN | 4116 | **0.0199±0.0084** | 580 | **0.0161±0.0071** |
| IDGNN | 89 | 0.0291±0.0069 | 75 | 0.0302±0.0022 |

(b) Runtime and performance comparison between fixed-point (FP) and bilevel (Bi) Methods.

| Runtime (s/win) | Fixed-point | Bilevel |
|---|---|---|
| Brain10 | 624 | **0.39** |
| PeMS04 | 0.72 | **0.049** |
| PeMS08 | 0.29 | **0.046** |
| England-COVID | 0.092 | **0.030** |
| Performance | Fixed-point | Bilevel |
| Brain10 | **94.7** | 94.5 |
| PeMS04 | 0.628 | **0.58** |
| PeMS08 | **0.501** | 0.56 |
| England-COVID | **2.97** | 3.05 |

## 5.3 CLASSIFICATION

We conducted classification experiments on Brain10, DBLP5, and Reddit4 datasets. Since these datasets consist of only one dynamic graph, we focused on testing the transductive case. Evaluation was done using the Area under the ROC Curve (AUC) metric. The average prediction AUC values and their corresponding standard deviations are presented in Table. 2. Our proposed model achieved the top rank in 2 out of 3 datasets and was the second best in the remaining dataset. These results demonstrate that our model successfully captures the long-range dependencies within the dynamic graphs, as reflected in the learned embeddings.

## 5.4 EFFICIENCY

We compare runtime and performance between naive gradient descent and bilevel optimization algorithms. To this end, we conduct comparisons on Brain10, England-COVID, PeMS04, and PeMS08. The results are summarized on the right. The results are computed by averaging the runtime of a whole epoch with the number of dynamic graphs $N$.

These methods have similar performance, but the runtime results show that the bilevel optimization algorithm is much faster than the naive gradient descent. Especially, in the Brain10 dataset, bilevel algorithm achieves *1600* times of speedup compared with naive gradient descent. Furthermore, we notice that the ratio of runtimes in PeMS04 and PeMS08 is $\frac{0.72}{0.29} = 2.48$, and the squared ratio of their number of nodes is $(\frac{307}{170})^2 = 3.26$. This confirms our complexity result for naive gradient descent, which is quadratic in terms of the number of nodes. On the other hand, the bilevel method exhibits only linear dependency. We also present the memory usage and runtimes of all methods on Reddit4 and DBLP5, but we leave those and the theoretical complexity comparison in the Appendix due to space limit. The memory efficiency of implicit models comes from the fact that implicit models can use few parameters and do not need to store the intermediate results. However, we need to store intermediate results and backpropagate for our bi-level method. Due to the simple RNN-free architecture of our method, our approach is competitive in runtime and memory. We provide a memory and runtime comparison on DBLP5 and Reddit4. The results are summarized in Tab. 3a.

## 6 CONCLUSIONS

In this paper, we propose a novel implicit graph neural network for dynamic graphs. As far as we know, this is the first implicit model on dynamic graphs. We demonstrate that the implicit model we proposed has the well-posedness characteristic. We proposed a standard optimization algorithm using the Implicit Function Theorem. However, the optimization turned out to be too computationally expensive for our model. Hence, we proposed a novel bilevel optimization algorithm to train our proposed model. We conducted extensive experiments on 6 real-world datasets and one toy dataset. The regression and classification tasks show that the proposed approach outperforms all the baselines in most settings. Finally, we also demonstrated that the proposed bilevel optimization algorithm obtains significant speedup over standard optimization while maintaining the same performance. A key limitation of our proposed approach is that only a single layer can be used for each time-stamp and it does not naturally lend itself to inductive setting. In the future, we plan on addressing this issue and also provide a diffusion model-based training algorithm.

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
