# OpenReview forum: "Implicit Neural Network on Dynamic Graphs"
_ICLR.cc/2024/Conference — Submitted to ICLR 2024_

### Official Review · Reviewer_qxmJ · 2023-10-26

**Soundness:** 2 fair
**Presentation:** 2 fair
**Contribution:** 2 fair
**Rating:** 5
**Confidence:** 3

**Summary:**

This paper proposes a new graph generative model for dynamic graphs based on implicit neural networks. The proposed method generalizes IGNN to the dynamic graphs, extending its capability to solve a broader range of problems. The well-posedness property has been shown for the proposed model. A bi-level optimization algorithm is developed for an efficient training of the proposed model. With the new training algorithm, the proposed model shows better performances on graph classification and regression tasks than baseline models.

**Strengths:**

- This is the first implicit model for dynamic graphs. The experimental results show that the implicit model for dynamic graphs can show better performances than non-implicit models.
- The proposed bi-level optimization algorithm can reduce the training time while having a competitive performance with the naive gradient descent algorithm.

**Weaknesses:**

- The paper proposes an implicit model for discrete-time *cyclic* dynamic graphs. I assume that the cyclic property is added to obtain the implicit representation of graphs, but the datasets used in the experiments do not have the cyclic property.
    - Moreover, I doubt that the performance of the synthetic experiments comes from the implicit representation. Since the representation at time step $T$ is directly related to the representation at time step $1$, through the learning (back-propagation) process, the model can directly utilize the information at time step $1$ to infer the class label at time step $T$. Hence, it is unclear whether the long-range dependency is captured correctly or not.
- The main theorem seems a direct consequence of Gu et al. (2020).
- The claimed 1600x speed-up seems like an overstatement. Although the proposed algorithm achieves a 1600x speed-up for the Brain10 dataset, the improvement is much lower for the other datasets. Having said that, I found that the improvement from the other datasets is not insignificant (10x improvement is also great).
    - Moreover, it would be much more meaningful if there were any analysis on why the algorithm performs well on the Brain10 dataset. What characteristics of the dataset lead to such an impressive performance increase?
- The representation of the manuscript can be improved further. Several notations are confusing, and a few terms are explained without having proper definitions. Here, I list some of them.
    - The notation $t$ is used for the depth of a layer and the time stamp of a graph (e.g., the first paragraphs of section 3). Although one may infer which t corresponds to which (based on location - superscript for layer and subscript for timestamp), it is difficult to follow the manuscript.
    - Transpose is denoted with superscript $T$, which is confusing with the timestamp T. Using \top latex command can alleviate the confusion.
    - Omega is not defined (Page 3, third line). I guess it means V
    - \ell in the equation on page 6 (where \nabla L(\omega) is defined) is not defined. So, I couldn’t follow the details after equation 8.
    - Please add references for datasets.
    - Use proper command for the citations. Use latex commands \citet and \citep for this.
    - Typo in the first sentence on Page 3 (l and d are both used to denote the dimension of the node attributes).
    - Typo in the matrix in Theorem 1 (the right-most column needs to be removed)

**Questions:**

- Why V in equation 2 is shared across time, and W is not?
- What makes the optimization ‘bi-level’? It would be better to have some additional background on the bi-level optimization methods.

---

> ### Author Response · Authors · 2023-11-21
>
> To Reviewer qxmj,
>
> We thank reviewer qxmj for the valuable feedback and recommendations for improving the manuscript.
>
> >	The paper proposes an implicit model for discrete-time cyclic dynamic graphs. I assume that the cyclic property is added to obtain the implicit representation of graphs, but the datasets used in the experiments do not have the cyclic property.
> >	Moreover, I doubt that the performance of the synthetic experiments comes from the implicit representation. Since the representation at time step T is directly related to the representation at time step 1, through the learning (back-propagation) process, the model can directly utilize the information at time step 1 to infer the class label at time step T. Hence, it is unclear whether the long-range dependency is captured correctly or not.
>
> Our model's development hinges on the cyclic property; however, as evidenced by experiments, it demonstrates efficacy on dynamic graphs in general. In response to the reviewer's feedback, we refined our synthetic experiments by shifting label information from time stamp 1 to time stamp 5. This adjustment ensures uniform difficulty in utilizing label information across all models. Implementing this change postpones our method towards achieving 100% accuracy by approximately 50 epochs. However, even with this modification, the baselines still struggle to fit the data. We put the plot in the updated appendix.
>
>
> >	The main theorem seems a direct consequence of Gu et al. (2020).
>
> The main theorem holds due to 1) the well-posedness result from Gu et al. (2020), as mentioned by the reviewer, and 2) the equivalent static representation of dynamic graphs. The contribution of our work lies in providing a framework to apply implicit graph neural network to dynamic graphs and proposing an efficient training algorithm for this problem. As we mentioned in the paper, using the naïve gradient descent (Gu et al. (2020)) to optimize this problem is extremely slow.
>
>
> >	The claimed 1600x speed-up seems like an overstatement. Although the proposed algorithm achieves a 1600x speed-up for the Brain10 dataset, the improvement is much lower for the other datasets. Having said that, I found that the improvement from the other datasets is not insignificant (10x improvement is also great).
> >	Moreover, it would be much more meaningful if there were any analysis on why the algorithm performs well on the Brain10 dataset. What characteristics of the dataset lead to such an impressive performance increase?
>
> The bilevel algorithm achieves 1600x speed-up on Brain10 because of the number of nodes. Brain10 has 5000 nodes. As we mentioned in the paper, naïve gradient descent has O(n^2) complexity per iteration, while bilevel method admits O(n) complexity. When the graph size is large, the speed-up grows quadratically. We also provided a detailed complexity analysis in the appendix.
>
> > The representation of the manuscript can be improved further. Several notations are confusing, and a few terms are explained without having proper definitions. Here, I list some of them.
>
>
>
> Thank you for pointing out, and we will fix them to make it more readable and clearer.
>
> > Why V in equation 2 is shared across time, and W is not?
>
> In fact, the variable V can be shared, leading to four distinct model configurations depending on whether both V and W are shared or not. It is crucial to emphasize that our convergence theorem is applicable across all these configurations. Our ablation study has revealed that the existing configuration is not only efficient but also competitive. However, it is noteworthy that sharing V may result in significant computational overhead, particularly considering the potential largeness of the original feature dimension.
>
> > What makes the optimization ‘bi-level’? It would be better to have some additional background on the bi-level optimization methods.
>
> In Eq (8), we have the upper lever problem (objective), and we also have lower lever problem (constraints). The variables ${z^{(i)}}$ are the optimal solutions to the lower lever problem. Such formulation produces a bilevel optimization problem. Thank you for your suggestion; we will introduce some bilevel optimization background in the related work.

---

### Official Review · Reviewer_NhSL · 2023-10-31

**Soundness:** 3 good
**Presentation:** 2 fair
**Contribution:** 3 good
**Rating:** 5
**Confidence:** 4

**Summary:**

This paper focuses on graph learning for dynamic graphs. As the oversmoothing issues and the failure to capture long-range dependencies are more severe on dynamic graphs, the authors propose an implicit graph neural network model to mitigate the issues. To remedy the computationally expensive training issue, they propose a single-loop training algorithm by changing the original optimization problem to a bi-level optimization problem. The experimental results on both classification and regression tasks show the superiority of the proposed model in terms of both performance and efficiency.

**Strengths:**

1. The idea of using implicit GNNs for dynamic graphs is sound and the motivation to mitigate the dilemma between capturing long-range dependencies and suffering from oversmoothing problems is reasonable and interesting.
2. The construction of the new equation for dynamic graphs in Eq (4) is good and the related theorems are sound.
3. The performance of synthetic experiments directly supports the claim that the proposed method can avoid over-smoothing and still be effective in capturing long-range dependencies.

**Weaknesses:**

1. To me, the relation between Lemma 2 and the relationship between Lemma 2 and Eq (8) is not very clear. In Lemma 2, how does $M_i$ get involved in the formula about $z_j$. Additionally, Eq (8) suggests that the new constraint is only about the last timestamp. In this case, is it necessary to have Lemma 2 to arrive at Eq (8)? Why cannot directly iterate Eq (5) to have $\phi(z, W, V; G_i)$. I would like to see more explanations regarding these.
2. The literature review may not be sufficient. As the paper focuses on implicit GNNs, I think the author may want to introduce and briefly discuss a few more recent implicit GNN works (e.g., CGS [1], EIGNN [2], USP [3]). Especially, USP seems to have a similar bilevel optimization problem, though it focuses on static graphs.
3. The descriptions for the experiments are not very clear. As mentioned in Table 3, the memory usage and the runtime are reported as per batch. But how batches are formed for a single graph? Randomly select some nodes or use some sampling methods (e.g., neighbor sampling)?

Overall, I think it's an interesting submission. But I hope the authors can clarify some questions I raise here.

References:

[1] Park, Junyoung, Jinhyun Choo, and Jinkyoo Park. Convergent graph solvers. ICLR 2022.

[2] Liu, Juncheng, Kenji Kawaguchi, Bryan Hooi, Yiwei Wang, and Xiaokui Xiao. Eignn: Efficient infinite-depth graph neural networks. NeurIPS 2021.

[3] Mingjie Li and Yifei Wang and Yisen Wang and Zhouchen Lin. Unbiased Stochastic Proximal Solver for Graph Neural Networks with Equilibrium States. ICLR 2023.

**Questions:**

1. Although the convergence guarantee is a good thing to see, I am just curious whether this is necessary to make the implicit graph model work well. Based on my understanding, existing implicit GNNs all have this property. In contrast, implicit models in other areas seem not always have this theoretical guarantee (e.g., DEQ [1] and MDEQ). They empirically work well.
2. Could you explain more about Hassian-vector Product as mentioned in the last paragraph of Sec 4? Can it be directly handled by a modern autodiff package? At least provide some reference materials in the appendix.

Minor ones:
1. There is no Table 3 caption. Please fix it.


References

[1] Deep Equilibrium Models. Shaojie Bai, J. Zico Kolter and Vladlen Koltun (NeurIPS 2019)

[2] Multiscale Deep Equilibrium Models. Shaojie Bai, Vladlen Koltun and J. Zico Kolter (NeurIPS 2020)

---

> ### Author Response · Authors · 2023-11-21
>
> To Reviewer NhSL,
>
> We thank reviewer NhSL for the valuable feedback and recommendations for improving the manuscript.
>
> > 1.	To me, the relation between Lemma 2 and the relationship between Lemma 2 and Eq (8) is not very clear. In Lemma 2, how does $M_i$
>  get involved in the formula about $z_j$
> . Additionally, Eq (8) suggests that the new constraint is only about the last timestamp. In this case, is it necessary to have Lemma 2 to arrive at Eq (8)? Why cannot directly iterate Eq (5) to have $\phi(z,W,V;G_i)$
> . I would like to see more explanations regarding these.
>
> In Lemma 2, we apply $T$ layers of GCN on $z_j$. When $j+k>T$, we let $M_{j+k}=M_i$ where $i=(i+k) mod T$. We will improve this Lemma and make it more readable.
>
> The purpose of lemma 2 is to develop an equivalence between the optimization problems in Equations (5) and (8). This allows us to demonstrate that a solution to the problem in Equation (5) is also a solution to the problem in Equation (8). As the reviewer has indicated, we can iterate Eq (5) to obtain function \phi. Lemma 2 serves as a rigorous certificate that such iterations do not change the problem.
>
> Please note that the constraint is applied to each time-stamp t in the range 1 to T in Equation (8). This constraint (and a similar constraint in (5) follows from Theorem 1, which ensures that the original problem in Equation (3) has a unique solution.
>
>
> 2.	The literature review may not be sufficient. As the paper focuses on implicit GNNs, I think the author may want to introduce and briefly discuss a few more recent implicit GNN works (e.g., CGS [1], EIGNN [2], USP [3]). Especially, USP seems to have a similar bilevel optimization problem, though it focuses on static graphs.
>
> Thank you. We will introduce these papers in the related work.
>
> > 3.	The descriptions for the experiments are not very clear. As mentioned in Table 3, the memory usage and the runtime are reported as per batch. But how batches are formed for a single graph? Randomly select some nodes or use some sampling methods (e.g., neighbor sampling)?
>
>
> A batch in our experiments consists of a collection of continuous timestamps (i.e., time stamps that fall within a fixed window). The window size is constant for all the approaches. Hence, the running time comparison is fair. We do not use any sampling methods.
>
>
> > 4.	Although the convergence guarantee is a good thing to see, I am just curious whether this is necessary to make the implicit graph model work well. Based on my understanding, existing implicit GNNs all have this property. In contrast, implicit models in other areas seem not always have this theoretical guarantee (e.g., DEQ [1] and MDEQ). They empirically work well.
>
> It is true DEQ and MDEQ work well empirically. We think this might be because their architecture makes them perform like contraction operators. We observe that, if we ignore the constraints on W, we can still converge under appropriate initialization in some datasets., but the model will diverge in other ones (producing meaningless embedding). Overall, we think the convergence guarantee is important to the implicit graph model.
>
>
> > 5.	Could you explain more about Hassian-vector Product as mentioned in the last paragraph of Sec 4? Can it be directly handled by a modern autodiff package? At least provide some reference materials in the appendix.
>
> To compute the product of Hessian and a vector: $Hv$, and $H=\frac{\partial^2 f}{\partial x^2}$. We compute the product by $Hv = \frac{\partial (\frac{\partial f}{\partial x })^T v }{\partial x}$. In this way, we are not explicitly computing the Hessian. Pytorch and other libraries support Hessian-vector Product directly: torch.autograd.functional.hvp().
>
> Here is a useful link illustrating the principles of Hessian–vector product:
> https://jax.readthedocs.io/en/latest/notebooks/autodiff_cookbook.html
>
> Thank you for your suggestion. We will add some references in the appendix.

---

### Official Review · Reviewer_gbnb · 2023-11-01

**Soundness:** 3 good
**Presentation:** 3 good
**Contribution:** 3 good
**Rating:** 6
**Confidence:** 3

**Summary:**

The paper addresses the limitations of graph convolution neural networks (GCNs) in capturing long-range dependencies and oversmoothing issues in dynamic graphs.

The authors propose IDGNN, a novel implicit neural network for dynamic graphs, which overcomes these issues and has a unique fixed point solution.

To efficiently train IDGNN, the authors pose an equivalent bi-level optimization problem and propose a single-loop training algorithm, achieving up to 1600x speed-up compared to the standard iterative algorithm.

Extensive experiments on real-world datasets demonstrate the superiority of IDGNN over state-of-the-art baseline approaches in both classification and regression tasks.

The paper also discusses the challenges in training implicit models and introduces an efficient bilevel optimization algorithm to overcome these challenges, resulting in improved computational efficiency during training.

The contributions of the paper include proving the existence of fixed-point representations in dynamic graphs, designing an implicit model for general dynamic graphs, and developing an efficient training algorithm for IDGNN.

**Strengths:**

Originality:

The paper introduces IDGNN, a novel implicit neural network for dynamic graphs, which addresses the limitations of existing graph convolution neural networks (GCNs) in capturing long-range dependencies and oversmoothing issues.
The authors propose a bi-level optimization framework and a single-loop training algorithm to efficiently train IDGNN, which is a novel approach in the context of dynamic graphs.

Quality:

The paper provides a rigorous analysis of the proposed IDGNN model, demonstrating its well-posedness and unique fixed point solution.
Extensive experiments on real-world datasets are conducted to evaluate the performance of IDGNN, comparing it to state-of-the-art baseline approaches.

Clarity:

The paper clearly presents the motivation, challenges, and contributions of the research.
The authors provide detailed derivations and explanations in the Appendix to support their claims and ensure clarity.

Significance:

The proposed IDGNN model and the efficient training algorithm have the potential to significantly improve the performance of dynamic graph neural networks, addressing the limitations of existing approaches.

The experimental results demonstrate the superiority of IDGNN over state-of-the-art baseline approaches in both classification and regression tasks, highlighting its practical significance.

**Weaknesses:**

The paper lacks a comprehensive discussion on the limitations of the proposed IDGNN model and the potential challenges in its practical implementation.

The experimental evaluation could be further strengthened by including more diverse and challenging datasets, as well as comparing the performance of IDGNN with a wider range of state-of-the-art approaches.

The paper could benefit from providing more insights into the interpretability of the IDGNN model and how it captures the underlying dynamics of the dynamic graphs.

The clarity of the paper could be improved by providing more intuitive explanations and visualizations of the proposed model and its training algorithm.

The paper could provide more details on the computational complexity and scalability of the proposed single-loop training algorithm, particularly in large-scale dynamic graph scenarios.

Overall, addressing these weaknesses would enhance the overall quality and impact of the paper.

**Questions:**

Can the authors provide more insights into the limitations of the IDGNN model and potential challenges in its practical implementation?

Could the authors consider including more diverse and challenging datasets in the experimental evaluation to further validate the performance of IDGNN?

It would be helpful if the authors could provide more details on the interpretability of the IDGNN model and how it captures the underlying dynamics of the dynamic graphs.

Can the authors clarify the computational complexity and scalability of the proposed single-loop training algorithm, particularly in large-scale dynamic graph scenarios?

Could the authors provide more intuitive explanations and visualizations of the proposed IDGNN model and its training algorithm to enhance the clarity of the paper?

It would be beneficial if the authors could discuss the potential applications and real-world use cases where IDGNN can be applied to address specific problems.

---

> ### Author Response · Authors · 2023-11-21
>
> To Reviewer gbnb,
>
> We thank reviewer gbnb for the valuable feedback and recommendations for improving the manuscript.
>
> > 1.	Can the authors provide more insights into the limitations of the IDGNN model and potential challenges in its practical implementation?
>
> We have mentioned some limitations of our method in the Conclusion section. The major limitation of our method is that it requires at least one GCN layer for each time stamp inside the window, which means picking a big window size can result in a large model. Another limitation is that predicting the current time stamp requires accessibility of all previous time stamps (within window size), which might not be practical in some cases since predictions might be required for future time stamp that is several days away.
>
> > 2.	Could the authors consider including more diverse and challenging datasets in the experimental evaluation to further validate the performance of IDGNN?
>
> As mentioned earlier, we believe we have added all standard benchmark datasets for node-level tasks (classification and regression) in dynamic networks. These datasets exhibit notable variations across domains, length, tasks, and scale, thereby ensuring a diverse and comprehensive coverage. It is worth highlighting that datasets specifically designed for discrete-time dynamic graphs in the context of node-level tasks are scarce [1]. Nevertheless, we have diligently curated and included the most prevalent ones in our analysis.
>
> [1] Xu, Da, et al. "Inductive representation learning on temporal graphs." arXiv preprint arXiv:2002.07962 (2020).
>
> > 3.	It would be helpful if the authors could provide more details on the interpretability of the IDGNN model and how it captures the underlying dynamics of the dynamic graphs.
>
> The main structure of our model is Z_{t+1}=\sigma(A_t Z_{t}W_t  + VX_t). In our design, Z_t carries out the dynamic information of the topology and flows through all the time stamps. In each time stamp, we inject the feature information as bias.
>
> > 4.	Can the authors clarify the computational complexity and scalability of the proposed single-loop training algorithm, particularly in large-scale dynamic graph scenarios?
>
> Since our method uses Hassian-Vector-Product to avoid explicitly computing the Hessian matrix, the main computational overheads of our method are the matrix multiplication during forward and backward processes. For each time stamp, we need to perform aggregation, which takes O(nd^2 + n^2d) or O(nd^2 + Ed), where E is the number of edges, and n is the number of nodes. In total, we have T time stamps, and the complexity for our method is O(Tnd^2 + Tn^2d). In a nutshell, our algorithm scales like GCN, which is a scalable method. Moreover, we have provided a detailed complexity comparison in the appendix.
>
> > 5.	Could the authors provide more intuitive explanations and visualizations of the proposed IDGNN model and its training algorithm to enhance the clarity of the paper?
>
> We have included a model diagram in the revised appendix. Kindly review it at your convenience.
>
> > 6.	It would be beneficial if the authors could discuss the potential applications and real-world use cases where IDGNN can be applied to address specific problems.
>
> Our model is designed for general node-level tasks and can be applied to many real-world cases. To name just a few, traffic accident predictions, epidemiological forecasting, bio-medical relationships etc.

---

### Official Review · Reviewer_mjoJ · 2023-11-08

**Soundness:** 2 fair
**Presentation:** 1 poor
**Contribution:** 2 fair
**Rating:** 3
**Confidence:** 4

**Summary:**

The paper presents IDGNN, an Implicit Neural Network for Dynamic Graphs, aimed at overcoming the limitations of graph convolution neural networks (GCNs), such as over-smoothing and the failure to capture long-range dependencies, especially in dynamic settings. The authors introduce a novel bilevel optimization framework for training IDGNN, which shows superior performance on real-world datasets in both classification and regression tasks compared to state-of-the-art approaches. They also demonstrate a significant speed-up in training times without compromising performance.

**Strengths:**

1. IDGNN is the first method to tackle the dynamic graph problem via an implicit neural network, filling a gap in the literature.
2. The model outperforms state-of-the-art methods on various real-world datasets, and the authors provide experimental validation.

**Weaknesses:**

1. The discussion about IGNN being able to avoid over-smoothing seems heuristic. IGNN ensures that the representation of the network is convergent, but it does not prevent over-smoothing problems.
2. The reasonableness of the assumption in Lemma 2 needs further explanation. For example, it says that Formula 3 has a unique embedding z, but which z in Formula 3 is referred to and under which conditions it is unique.
3. In Lemma 2, ``let W_{j+k} denote M_{i}``. needs further explanation.
4. Due to the question regarding Lemma 2, I am unable to determine the reasonableness of bilevel problem (8). (8) utilizes multi-block bilevel optimization for solving, and when solving (8), the paper makes extensive use of approximations without explaining their validity or drawbacks. Additionally, the writing of the section on training IDGNN is poor, and there is insufficient clarity in comparing it with existing training methods.
5. Lack of experimental results on common datasets, such as QM9 and TUdataset.

**Questions:**

See weakness

---

> ### Author Response · Authors · 2023-11-21
> **Rebuttal 1/2**
>
> To Reviewer mjoJ,
>
> We thank the reviewer for your helpful feedback on the manuscript. Please find our detailed response below:
>
>
> > 1. The discussion about IGNN being able to avoid over-smoothing seems heuristic. IGNN ensures that the representation of the network is convergent, but it does not prevent over-smoothing problems.
>
>
> Implicit graph neural networks have been able to capture long-range dependency while alleviating over-smoothing issues [1] [2] [3]. While there is no theoretical guarantee that the converged implicit representations necessarily avoid over-smoothing issues (as pointed out by the reviewer), there is enough empirical evidence (including in our experiments) that the implicit graph neural networks help alleviate these.
>
> [1] Liu, Juncheng, et al. "Eignn: Efficient infinite-depth graph neural networks." Advances in Neural Information Processing Systems 34 (2021): 18762-18773.
> [2] Chen, Qi, et al. "Optimization-induced graph implicit nonlinear diffusion." International Conference on Machine Learning. PMLR, 2022.
> [3] Yang, Yongyi, et al. "Graph neural networks inspired by classical iterative algorithms." International Conference on Machine Learning. PMLR, 2021.
>
>
> > 2. The reasonableness of the assumption in Lemma 2 needs further explanation. For example, it says that Formula 3 has a unique embedding z, but which z in Formula 3 is referred to and under which conditions it is unique.
>
>
> Lemma 2 holds for any vector z that satisfies Equation 3 and Equation 3 is guaranteed a unique solution if the condition in Theorem 1 is satisfied (which we ensure during training). Please note that Lemma 2's purpose is to show equivalence between the problem in Equation 5 and the one in Equation 8 by demonstrating that the constraints imply the same fixed-point solution.  We will update Lemma 2 and make it more readable.
>
>
> > 3. In Lemma 2, let W_{j+k} denote M_{i}. needs further explanation
>
>
> We believe the reviewer is referring to the line let M_{j+k} denote M_{i} : i=(j+k) mod T.  Our goal here was to define M_{j+k}=M_{i} when j+k > T, since M_{i} is only defined for i in {1,…,T}. We realize there is a better way to express this: defining the mod operator \Bar{j+k} =(j+k) mod T. We will update Lemma 2 and make it more readable.
>
>
> > 4.	Due to the question regarding Lemma 2, I am unable to determine the reasonableness of bilevel problem (8). (8) utilizes multi-block bilevel optimization for solving, and when solving (8), the paper makes extensive use of approximations without explaining their validity or drawbacks. Additionally, the writing of the section on training IDGNN is poor, and there is insufficient clarity in comparing it with existing training methods.
>
>
> We have given a justification about how our bilevel algorithm approximates the hypergradient. The goal of most gradient-based bilevel optimization algorithms is to accurately and rapidly estimate the hypergradient [1]. The difficulties in obtaining the hypergradient are: 1) the estimation of inverse Hessian matrix of lower-level problem, and 2) the accessibility to the optimal lower-level solution. Inspired by [2], we use moving average estimator for approximating the lower-level solution and the inverse Hessian-vector product.
>
> In cases where the lower-level problem is strongly convex, the errors introduced by these approximations are well-contained, aligning with the approach in [2] and enabling optimal sample complexity. However, when dealing with a multi-block non-convex lower-level problem, as expressed in Eq (8), the algorithm presented in [2] is not directly applicable. Nevertheless, it is crucial to note that the optimal solution to our lower-level problem corresponds to the fixed point of Eq (3), as per Lemma 2. Leveraging this insight, we employ a fixed-point iteration to update the lower-level solution. Empirical results strongly support the efficacy of our approximations in practice.
>
> [1] Liu, Bo, et al. "Bome! bilevel optimization made easy: A simple first-order approach." Advances in Neural Information Processing Systems 35 (2022): 17248-17262.
> [2] Hu, Quanqi, Yongjian Zhong, and Tianbao Yang. "Multi-block min-max bilevel optimization with applications in multi-task deep auc maximization." Advances in Neural Information Processing Systems 35 (2022): 29552-29565.

---

> ### Author Response · Authors · 2023-11-21
> **Rebuttal 2/2**
>
> > 5. Lack of experimental results on common datasets, such as QM9 and TUdataset.
>
>
> Our paper focuses on node-level tasks on dynamic graphs. The QM9 is a static graphs dataset, and TUdataset only contains dynamic graphs for graph classification. Moreover, we believe that the datasets we used are common datasets in field of dynamic graphs learning as these were selected from influential papers [1] [2] on dynamic graphs. As indicated by [3], Publicly available datasets for node classification in the dynamic setting are rare. We have tried to cover a diverse collection of them.
>
> [1] Gao, Jianfei, and Bruno Ribeiro. "On the equivalence between temporal and static equivariant graph representations." International Conference on Machine Learning. PMLR, 2022.
> [2] Xu, Da, et al. "Inductive representation learning on temporal graphs." arXiv preprint arXiv:2002.07962 (2020).
> [3] Pareja, Aldo, et al. "Evolvegcn: Evolving graph convolutional networks for dynamic graphs." Proceedings of the AAAI conference on artificial intelligence. Vol. 34. No. 04. 2020.

---

### Meta-Review · Area_Chair_s3sn · 2023-12-07

**Metareview:**

This paper introduces IDGNN, an implicit neural network for dynamic graphs that addresses the issues of oversmoothing and long-range dependencies. The authors propose a bi-level optimization problem and a single-loop training algorithm to efficiently train the model, achieving up to 1600x speed-up compared to the standard iterative algorithm. Extensive experiments on real-world datasets show that IDGNN outperforms state-of-the-art baseline approaches in both classification and regression tasks.

Strength
- IDGNN is the first implicit neural network method for dynamic graphs, effectively mitigates oversmoothing problems and captures long-range dependencies, outperforming state-of-the-art methods on various real-world datasets.

Weakness
- The method claims to solve the over-smoothing problem, but there is no theoretical support. The rebuttal also confirms this point.
- Reviewers show concerns on Lemma 2. More clarification needs to be included.
- The experimental evaluation could be further strengthened by including more diverse and challenging datasets, as well as comparing the performance of IDGNN with a wider range of state-of-the-art approaches.

**Justification For Why Not Higher Score:**

Please see weakness above.

While authors have offered material for the theoretical and empirical part, the submission needs a significant revision to meet the acceptance criterion.

**Justification For Why Not Lower Score:**

N/A.

---

### Decision · Program_Chairs · 2024-01-16

Reject